# Aromatized liposomes for sustained drug delivery

Yang Li[1], Tianjiao Ji[1], Matthew Torre[2], Rachelle Shao[1], Yueqin Zheng[1], Dali Wang[1], Xiyu Li[1], Andong Liu[1], Wei Zhang[1], Xiaoran Deng[1], Ran Yan[1] & Daniel S. Kohane[1] ✉

Insufficient drug loading and leakage of payload remain major challenges in designing liposome-based drug delivery systems. These phenomena can limit duration of effect and cause toxicity. Targeting the rate-limiting step in drug release from liposomes, we modify (aromatized) them to have aromatic groups within their lipid bilayers. Aromatized liposomes are designed with synthetic phospholipids with aromatic groups covalently conjugated onto acyl chains. The optimized aromatized liposome increases drug loading and significantly decreases the burst release of a broad range of payloads (small molecules and macromolecules, different degrees of hydrophilicity) and extends their duration of release. Aromatized liposomes encapsulating the anesthetic tetrodotoxin (TTX) achieve markedly prolonged effect and decreased toxicity in an application where liposomes are used clinically: local anesthesia, even though TTX is a hydrophilic small molecule which is typically difficult to encapsulate. Aromatization of lipid bilayers can improve the performance of liposomal drug delivery systems.

Liposomes have been explored for drug delivery applications for more than 50 years[1]. Liposomal formulations have been approved for clinical use in treating various diseases including cancer, fungal infections and pain[2]. Despite this success, some basic challenges remain in encapsulating and controlling the release of payloads[3], particularly with hydrophilic small molecule drugs[4]. Considerable drug release can occur immediately following administration[4,5], which may lead to systemic or local toxicity[6].

There are emerging efforts in developing synthetic lipids with chemical modifications in headgroups[7], linkers[8] or hydrophobic domains[9,10] to provide control over the intermolecular forces, phase preference, and macroscopic behavior of liposomes[11]. Movement across lipid bilayers is the rate-limiting step in the passive diffusion of molecules through cellular membranes and artificial membranes such as those of liposomes, because the hydrophobic phase (lipid bilayer) is 100–1000 times more viscous than the surrounding aqueous phase[12].

Inspired by the transportation of molecules across cell membranes, we reasoned that engineering the lipid bilayers of liposomes via chemical modification may improve the stability of liposomes and prolong the release of payloads.

Here, we report a strategy where we chemically modify the lipid bilayers with synthetic phospholipids whose acyl chains were covalently bound to a terminal aromatic group, creating aromatized liposomes. We studied the effect of liposome aromatization on drug encapsulation and release. We used the ultrapotent local anesthetic tetrodotoxin (TTX) as the model drug, as it has many of the properties that are most problematic in encapsulation in liposomes: low molecular weight, high hydrophilicity, and the potential for marked toxicity from uncontrolled release. We performed a proof-of-concept study demonstrating the in vivo effect of liposomal aromatization in local anesthesia, an application where liposomes are used clinically. Since the liposomes were intended for local administration and effect, they

[1]Laboratory for Biomaterials and Drug Delivery, Department of Anesthesiology, Division of Critical Care Medicine, Children's Hospital Boston, Harvard Medical School, Boston, MA 02115, US. [2]Department of Pathology, Brigham and Women's Hospital, Boston, MA 02115, US. ✉e-mail: daniel.kohane@childrens.harvard.edu

were designed to be larger than nanoscale liposomes intended for systemic delivery, to extend duration of release[13], and slow loss of particles from the site of injection[14].

# Results

## Synthesis of acyl chain-modified phospholipids

To examine the hypothesis that aromatized liposomes could enhance sustained drug release (Fig. 1a), we coupled the commercially available lipid 1-palmitoyl-2-hydroxy-sn-glycero-3-phosphocholine (lyso-PC) to a fatty acid bearing terminal aromatic group (aromatized fatty acid; Fig. 1b). We selected phenyl (one aromatic ring) and coumarin (two rings) groups, as their small size would make them less likely to interfere with the packing of hydrocarbon chains of phospholipids.

The aromatized fatty acid was synthesized via substitution of an aromatic group with a corresponding brominated fatty acid. Methyl 16-bromohexadecanoate, phenol, and potassium carbonate were mixed in anhydrous acetonitrile at 60 °C overnight to create methyl 16-phenoxyhexadecanoate. Deprotection of the carboxylic acid yielded 16-phenoxy-palmitic acid, which was then reacted with 1-palmitoyl-2-hydroxy-sn-glycero-3-phosphocholine (lysoPC) to create phenoxy-conjugated dipalmitoylphosphatidylcholine (Ph-DPPC). In the $^1$H-NMR spectrum of Ph-DPPC, the representative signals of aromatic rings at 7.2 and 6.9 ppm clearly demonstrated the successful conjugation of a phenoxy group to an acyl chain of the phospholipid (Supplementary Figs. 1–4), which was also confirmed by liquid chromatography–mass spectrometry (LC-MS, Fig. 1c). A combination

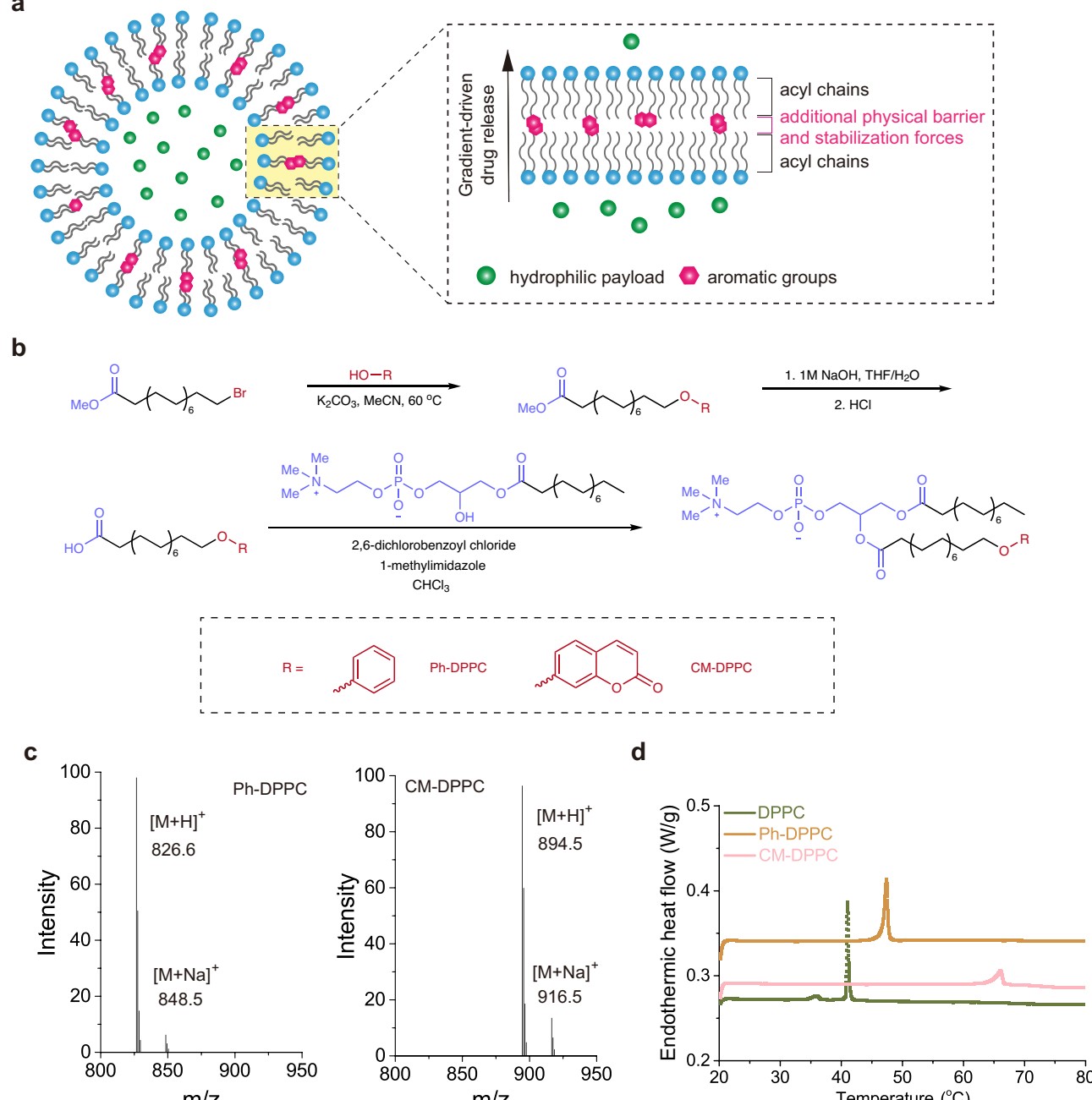

**Fig. 1 | Design of aromatized liposomes for sustained drug delivery. a** Schematic of aromatized liposomes for sustained drug delivery. **b** Synthesis of phenoxy-conjugated phospholipid (Ph-DPPC) and coumarin-conjugated phospholipid (CM-DPPC). **c** LC–MS characterization of Ph-DPPC and CM-DPPC. **d** Differential scanning calorimetry (DSC) study of liposomes prepared with 100% DPPC (Green), Ph-DPPC (Orange), and CM-DPPC (Pink).

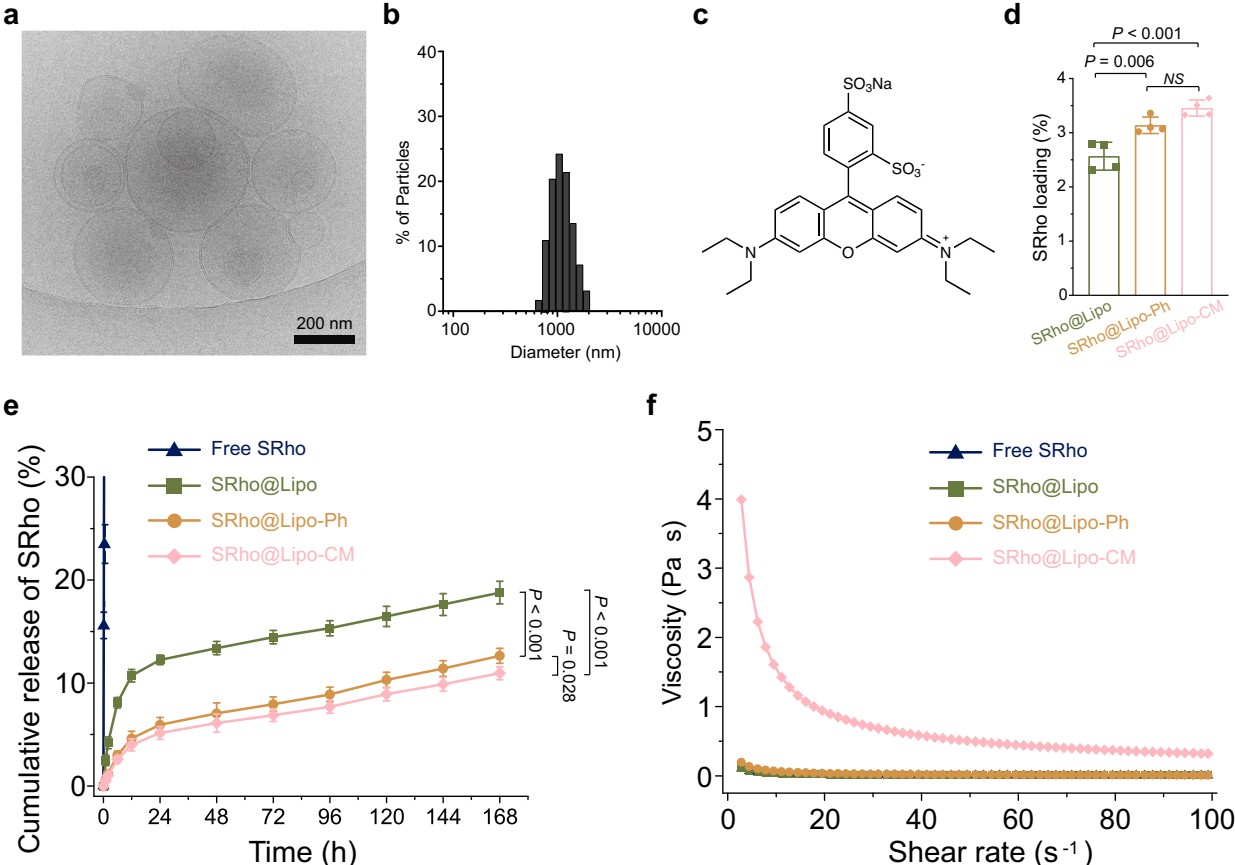

**Fig. 2 | Physicochemical characterizations of aromatized liposomes.**
**a** Cryogenic electron micrograph of aromatized liposome. Scale bar: 200 nm. The experiment was repeated with 4 independent samples, yielding similar results. **b** Diameters of aromatized liposome measured by dynamic light scattering. The experiment was repeated with 4 independent samples, yielding similar results. **c** Structure of Sulforhodamine B (SRho). **d** SRho loading in different chemically modified liposomes. Data are presented as mean ± SD, $n = 4$ independent experiments. Statistical analysis was performed using one-way ANOVA with a Tukey post hoc test. SRho@Lipo vs SRho@Lipo-Ph, $P = 0.0061$. SRho@Lipo vs SRho@Lipo-CM, $P = 0.0003$. SRho@Lipo-Ph vs SRho@Lipo-CM, $P = 0.1031$. **e** Cumulative

release of SRho from different formulations at 37 °C. Data are presented as mean ± SD, $n = 4$ independent experiments. Statistical analysis was performed using one-way ANOVA with a Tukey post hoc test. $P$-values compare groups at 168 h. SRho@Lipo vs SRho@Lipo-Ph, $P < 0.0001$. SRho@Lipo vs SRho@Lipo-CM, $P < 0.0001$. SRho@Lipo-Ph vs SRho@Lipo-CM, $P = 0.0279$. *$P < 0.05$. **$P < 0.01$. ***$P < 0.001$. **f** Viscosity of different liposomal formulations. In **d**–**f**, navy triangle, green square, orange circle and pink diamond represent free SRho, liposome encapsulating SRho (SRho@Lipo), phenyl-modified liposome encapsulating SRho (SRho@Lipo-Ph), and Coumarin-modified liposomes encapsulating SRho (SRho@Lipo-CM), respectively.

of 2,6-dichlorobezoyl chloride and 1-methylimidazole was selected as the coupling reagent for the synthesis due to their high potency in activating fatty acids[15], which lead to the coupling reaction being completed within 12 h in high yield (>75%). The coumarin-conjugated phospholipid (CM-DPPC, Supplementary Fig. 5) was synthesized following the same route. Coupling of a phenoxy and coumarin group increased the phase transition temperature of DPPC from 41 °C to 48 and 66 °C, respectively (Fig. 1d).

**Preparation of aromatized liposomes and in vitro evaluation**
Given that local anesthesia is for local (nerve) not systemic delivery, the liposomes were designed to be comparatively large[13,16,17]. Aromatized liposomes were prepared via thin-film hydration followed by agitation and freeze-thaw method to produce microscale liposomes[18,19], which favors local action following local administration[13]. A dried lipid thin film containing Ph-DPPC, 1,2-dioleoyl-sn-glycero-3-phosphocholine (DOPC), 1,2-distearoyl-sn-glycero-3-phosphatidylglycerol (DSPG), and cholesterol was hydrated with phosphate buffered saline (PBS). A combination of phosphatidylcholine (PC), phosphatidylglycerol (PG) and cholesterol is used in FDA approved liposome products for local administration. DOPC was used as in clinically used liposomes (like DepoDur) for local injection. DSPG was used because we found that negatively charged DSPG can enhance

the encapsulation of positively charged TTX[18]. Mechanical agitation was applied during hydration to facilitate membrane fusion and formation of multivesicular vesicles[7]. Liposomes made with Ph-DPPC and natural phospholipids were a mixture of unilamellar and multivesicular spherical vesicles as confirmed by cryogenic electron microscopy (Cryo-EM) (Fig. 2a), with a volume-weighted diameter of 1.05 ± 0.10 μm (Fig. 2b). The liposomes displayed similar size and did not aggregate after storage at 4 °C for several weeks, indicating good stability (Supplementary Fig. 6a, b). 100 nm liposomes could be produced after extrusion through polycarbonate filters (Supplementary Fig 6c, d); nanoscale is favorable for systemic administration. Release kinetics confirmed that in nanoscale liposomes, aromatization also increased drug loading and decreased the release rate of SRho (Supplementary Fig 6e, f)

The hydrophilic small molecule fluorophore Sulforhodamine B (SRho, $\log P = -0.53$, Fig. 2c)[18] was loaded into liposomes by hydrating the lipid cake with a 10 mg/mL SRho solution in PBS. The loading of SRho in aromatized liposomes (SRho@Lipo-Ph) was 19% greater than in unmodified liposomes (Fig. 2d), showing slightly increased drug loading efficiency as a result of aromatization. The release kinetics of SRho was evaluated in vitro. Free (unencapsulated) dye was released within 6 h. In SRho@Lipo, more than 12% SRho was released in 24 h. In comparison, less than 6.0% and 5.5% of SRho was released in 24 h from

SRho@Lipo-Ph and SRho@Lipo-CM (Fig. 2e). Compared to Lipo-Ph, Lipo-CM did not considerably increase the drug loading or reduce the cumulative drug release in first 24 h. Viscosity is an important factor for injectability. The viscosity of different formulations was characterized at a range of angular frequencies. The aqueous solution of Lipo-CM was notably more viscous than Lipo-Ph (Fig. 2f). Thus, we chose Lipo-Ph for the subsequence studies.

We further modified the acyl chain of phospholipids with dibenzocyclooctyne (DBCO) (DBCO-DPPC, Supplementary Fig. 7a, b), a group with three rings including two aromatic rings. The loading of SRho in DBCO-modified liposomes was $2.3 \pm 0.3\%$, comparable to that in unmodified liposomes (Supplementary Fig. 7c). 21% of SRho was released from SRho@Lipo-DBCO in 24 h, which was faster than from SRho@Lipo (Supplementary Fig. 7d). The bulky DBCO group may interfere with the packing of hydrocarbon chains, making the liposome leakier[20,21].

Since the covalently conjugated phenyl group slowed the release of payload, we studied the extent to which physical (non-covalent) encapsulation of phenol into lipid bilayers would stabilize liposomes. To compare the effects of covalent conjugation and physical encapsulation, we co-encapsulated SRho with 1 mM phenol into liposomes (Ph+SRho@Lipo) (Supplementary Fig. 8). Ph+SRho@Lipo released 7.1% of SRho in 24 h, 42% less than from unmodified liposomes, but 20% more than from covalently aromatized liposomes.

Phenol is intrinsically toxic[22,23], as confirmed by cytotoxicity data (Supplementary Fig. 9). Therefore, in order to be able to assess the effect of non-covalently incorporated aromatic groups in vivo, we co-encapsulated SRho with 1 mM indocyanine green (ICG)[24], an FDA approved fluorophore containing four aromatic rings (Supplementary Fig. 8). SRho release from ICG+SRho@Lipo was less rapid than from unmodified liposomes, but more than from covalently aromatized liposomes. Since there was no statistically significant difference between the effect of Ph and ICG on drug release, we use the latter in downstream experiments.

## Effect of liposome aromatization on different payloads

We expanded our study of the effect of liposome aromatization to include payload molecules with different characteristics such as hydrophilicity and molecular weight. The hydrophilicity of different payloads was measured by their octanol-water partition coefficients (LogP, Supplementary Fig. 10). In addition to SRho described above, we encapsulated the following small molecule compounds (Fig. 3a), using the same encapsulation and analytical techniques: tetrodotoxin (TTX, logP = 0.13), a hydrophilic ultrapotent local anesthetic; bupivacaine hydrochloride (Bup, logP = 1.24), an amphiphilic amino-amide local anesthetic in current clinical use; doxorubicin hydrochloride (Dox, logP = 0.33), a chemotherapeutic drug that has been used clinically in liposomes to treat cancer. These are logPs at pH 7.4, the pH at which the compounds were encapsulated.

Loading of small compounds was increased by 19–60% by aromatization of liposomes (Supplementary Table 1). Although molecular weight did not correlate strongly with the degree of increase of drug loading ($R^2 = 0.32$), hydrophilicity showed good correlation with the degree of increase ($R^2 = 0.92$; Supplementary Fig. 11). Release in the first 24 h was reduced by 30–60% by aromatization of liposomes (Fig. 3b–d, Supplementary Table 2). Neither molecular weight (Fig. 3e; $R^2 = 0.01$) nor hydrophilicity (Fig. 3f; $R^2 = 0.00$) showed correlation with the degree of reduction.

Similar loading and release experiments were done with liposomes loaded with the macromolecules rhodamine-conjugated polyethylene glycol (1 kDa and 10 kDa, LogP = −2.05 and −2.19, respectively) and fluorescein isothiocyanate-conjugated albumin (LogP = −2.39, Supplementary Fig. 10). Although aromatization had a modest (or no) effect on drug loading of macromolecules (Supplementary Table 1), aromatization decreased the release of macromolecules in the first 24 h from liposomes (Fig. 3g–i). The effect of liposomal aromatization on drug release increased with increasing molecular weight (Fig. 3j; $R^2 = 0.81$) and increasing hydrophilicity (Fig. 3k; $R^2 = 0.97$).

These data showed that aromatized liposomes had benefits over conventional liposomes for delivery of a wide range of small and large molecules.

In anticipation of in vivo experimentation, TTX was also co-encapsulated in unmodified liposomes with ICG (ICG + TTX@Lipo). 13% of TTX was released from ICG + TTX@Lipo within 24 h, which was less rapid than from unmodified liposomes, but more than from aromatized liposomes in the same period (Supplementary Fig. 12).

## Nerve blockade and systemic toxicity

To assess whether the prolongation of release and reduction in drug release with aromatized liposomes could translate into enhanced duration of effect and reduced toxicity in vivo, we used them in a rat model of peripheral nerve block. Aromatic (TTX@Lipo-Ph) and unmodified liposomes (TTX@Lipo) with or without TTX showed similar volume-weighted diameters (Supplementary Fig. 13a and Table 3). All liposomes had zeta potentials around −30 mV, indicating that the incorporation of aromatized phospholipids or encapsulation of TTX did not change their surface charge (Supplementary Fig. 13b). The low viscosity of all liposomal formulations (≤100 mPa s; Supplementary Fig. 13c) indicated that they were syringe injectable.

Rats were injected at the left sciatic nerve with 300 µL of different TTX formulations (Fig. 4a). Local anesthesia was assessed by a modified hotplate test[25,26] in which thermal latency, the time in seconds a rat left its hindpaws on a hotplate, was measured in both hindpaws (2 s was the baseline, and 12 s was maximal). The duration of deficits in the injected (left) hindpaw reflected the duration of sensory nerve block, whereas deficits in the contralateral uninjected (right) hindpaw indicated effects from systemic distribution of TTX.

Formulations were delivered locally, by injection directly at the nerve and had their principal therapeutic effect at that local site. The duration of nerve block reflected the kinetics of effective levels of drug at the site of injection. Similarly, the contralateral block is a reflection of systemic drug levels (i.e. levels in blood). Changes in the latencies and durations of contralateral nerve block are therefore metrics of the effect of formulational changes on pharmacokinetics.

The starting point for dose selection for free TTX in the in vivo study was based on dose-response curves that we have reported[26], where 4–5 µg of TTX produced 1.5–2.5 h of sciatic nerve blockade, with a shorter duration of deficits in the contralateral extremity. The starting dose of TTX in unmodified liposomes was adapted from our previous work[18], where 22 µg of TTX in liposomes produced effective sciatic nerve blockade lasting 17.2 h. A similar dose was used in the modified liposomes. Subsequent increases in dose were studied by incremental increases in dose-response curves, with dose escalation until systemic toxicity became dose-limiting.

Duration of block in rats injected at the sciatic nerve with free TTX increased with increasing dose (Fig. 4b), but so did systemic toxicity (Fig. 4c, d). 4 µg of TTX at the sciatic nerve resulted in a mean duration of sensory nerve block of $2.5 \pm 0.6$ h (Fig. 4b), with contralateral block lasting $2.1 \pm 0.5$ h. That dose of free TTX resulted in contralateral blocks in all animals (Fig. 4c), indicating systemic toxicity. Injection of 5 µg of free TTX was uniformly fatal (Fig. 4d).

Encapsulation of TTX in liposomes decreased systemic toxicity substantially[27], increasing the TTX dose that could be delivered. (The actual dose in each formulation varied slightly due to differences in TTX loading efficiency; Supplementary Table 4). Nerve block from TTX@Lipo containing 20.4 µg TTX lasted $19.9 \pm 4.4$ h (Fig. 4b). The contralateral block occurred in 62.5% of animals (Fig. 4c) with a duration of $0.8 \pm 0.3$ h, a peak latency of $7.5 \pm 1.0$ s, and there were no

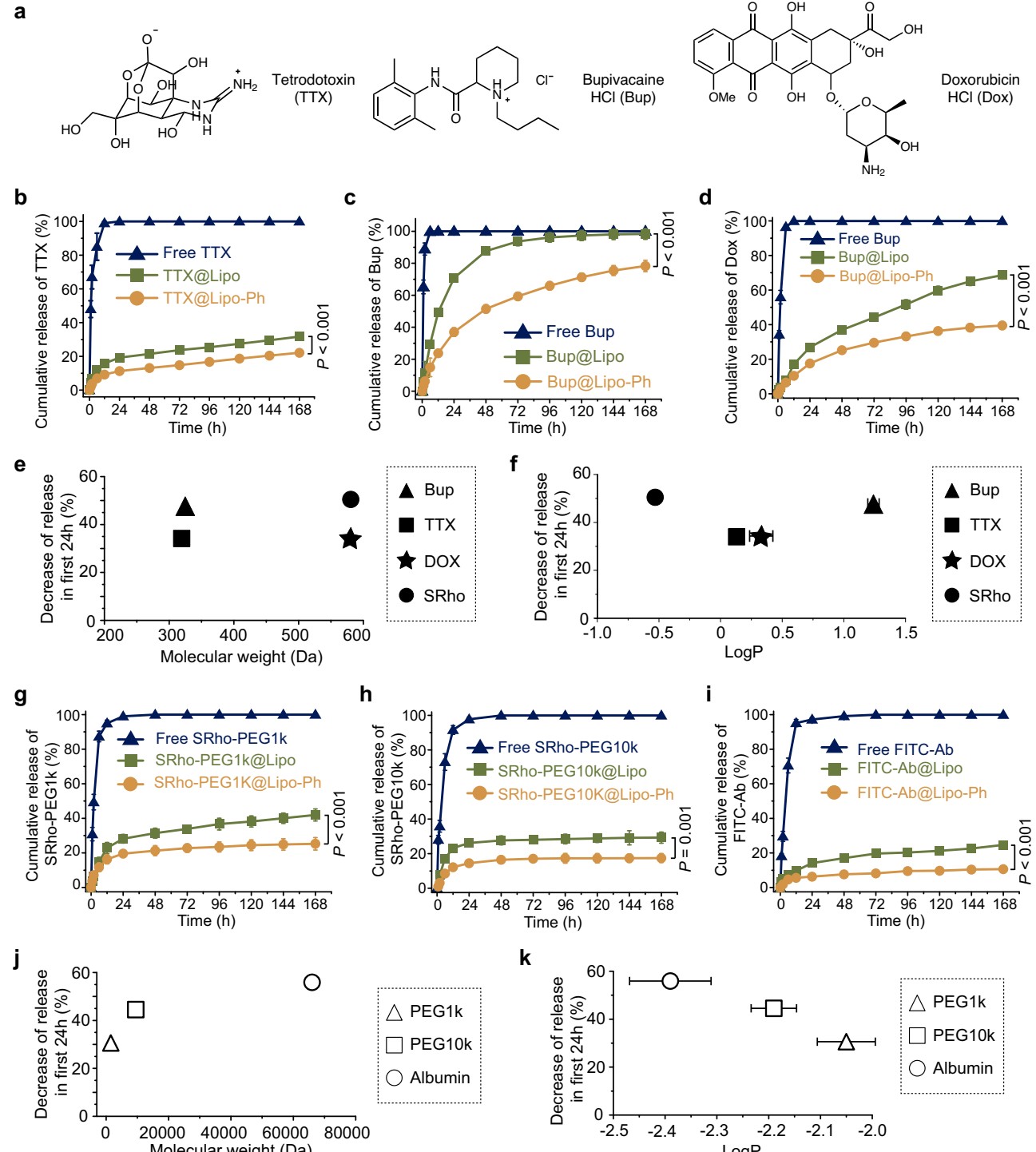

**Fig. 3 | Effect of liposome aromatization on different payloads. a** Structures of different small molecular payloads. **b−d** Cumulative release of Tetrodotoxin (TTX, **b**), Bupivacaine hydrochloride (Bup, **c**), Doxorubicin hydrochloride (Dox, **d**). **e**, **f** The decrease in the release of small molecular payloads as the function of molecular weight (**e**) and hydrophilicity (**f**). Cumulative release of rhodamine-conjugated polyethylene glycol with a molecular weight of 1000 (SRho-PEG1k, **g**), rhodamine-conjugated polyethylene glycol with a molecular weight of 10,000 (SRho-PEG10k, **h**), albumin-fluorescein isothiocyanate conjugate (FITC-Ab, **i**). **j**, **k** The decrease in drug release as the function of molecular weight (**j**) and hydrophilicity (**k**). In **b−d** and **g−i**, navy triangle, green square, and orange circle represent free payloads, liposomes encapsulating different payloads and

aromatized liposomes encapsulating different payloads, respectively. In **b−d**, **f**, **g−i**, **k**, data are presented as mean ± SD, $n = 4$ independent experiments, and statistical analysis was performed using one-way ANOVA with a Tukey post hoc test. In **b−d** and **g−i**, $P$-values compare groups at 168 h. TTX@Lipo vs TTX@Lipo-Ph, $P < 0.0001$; Bup@Lipo vs Bup@Lipo-Ph, $P < 0.0001$; Dox@Lipo vs Dox@Lipo-Ph, $P < 0.0001$; SRho-PEG1k@Lipo vs SRho-PEG1k@Lipo-Ph, $P < 0.0001$; SRho-10k@Lipo vs SRho-10k@Lipo-Ph, $P = 0.0006$; FITC-Ab@Lipo vs FITC-Ab@Lipo-Ph. $P < 0.0001$. In **f**, Bup vs Dox, $P < 0.0001$; Dox vs TTX, $P = 0.0020$; TTX vs SRho, $P < 0.0001$. In **k**, PEG1k-SRho vs PEG10k-SRho, $P = 0.0346$; PEG1k-SRho vs FITC-Ab, $P < 0.0001$; PEG10k-SRho vs FITC-Ab, $P = 0.0057$. *$P < 0.05$, **$P < 0.01$. ***$P < 0.001$.

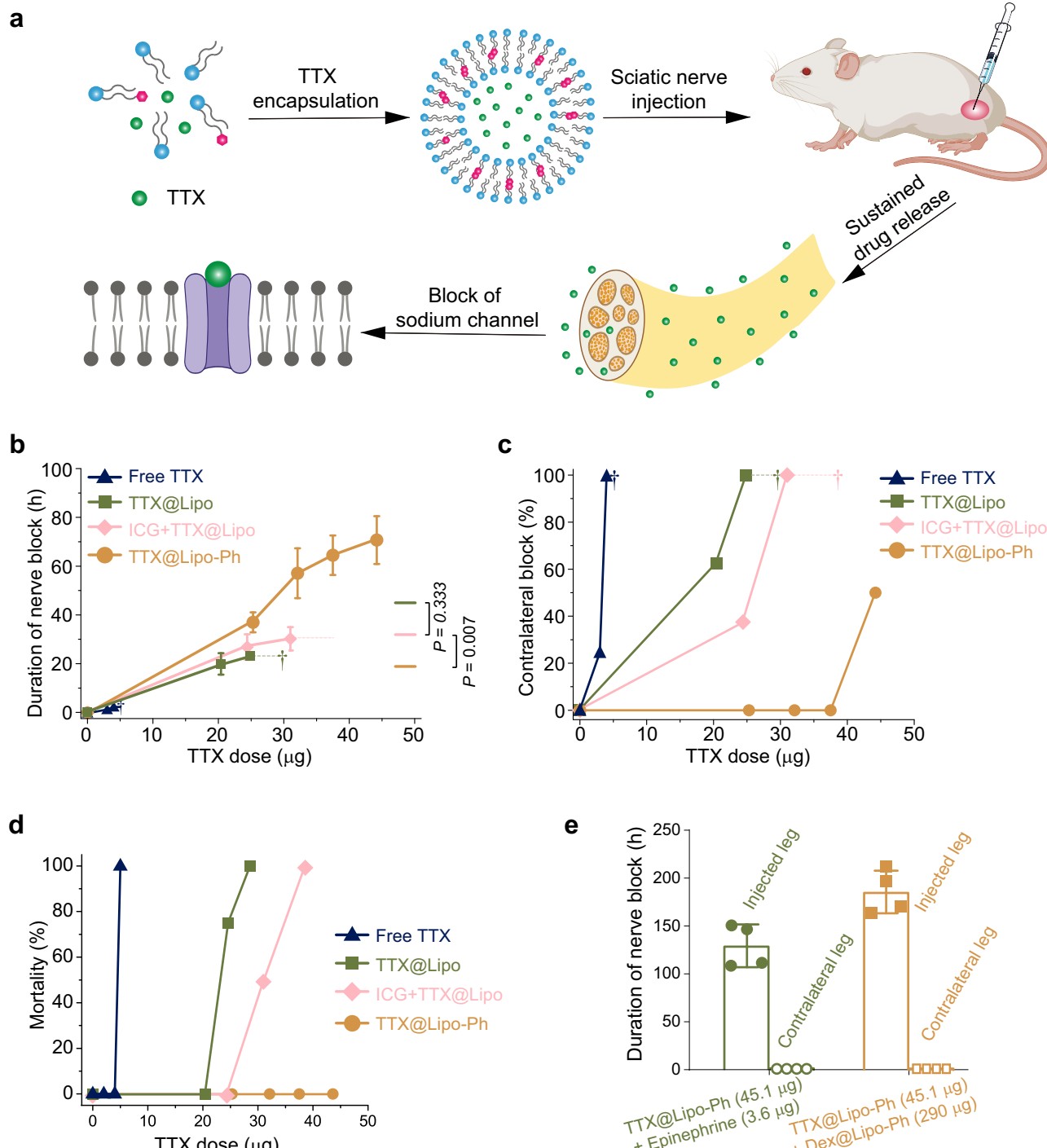

**Fig. 4 | In vivo performance of TTX formulations. a** Schematic of liposomes encapsulating tetrodotoxin, injection at the sciatic nerve, and prolonged duration local anesthesia due to blockade of voltage-gated sodium channels. **b** Duration of sensory nerve blockade from different formulations injected at the sciatic nerve. Daggers indicate 100% mortality. Data are presented as means ± SD. $n = 4$ biologically independent animals. *P*-values are from unpaired two-tailed *t*-test. $P = 0.333$ comparing TTX@Lipo at 24.8 μg TTX and ICG + TTX@Lipo at 24.4 μg; ***P* = 0.007 comparing ICG + TTX@Lipo at 31.0 μg TTX and TTX@Lipo-Ph at 32.1 μg TTX. **c** Frequency of block in the contralateral (uninjected) leg. **d** Mortality from different

formulations. Deaths occurred within the first 0.5 h after injection in animals treated with free TTX, within 0.5–1.0 h in animals treated with TTX@Lipo, and within 0.7–1.5 h in animals injected with TTX + ICG@Lipo. In **b**–**d**, navy triangle, green square, pink diamond, and orange circle represent free TTX, liposomes encapsulating TTX (TTX@Lipo), liposomes encapsulating TTX and ICG (ICG + TTX@Lipo) and aromatized liposomes encapsulating TTX (TTX@Lipo-Ph), respectively. **e** Effect of epinephrine (green circle) and dexamethasone (orange square) on the duration of nerve block from TTX@Lipo-Ph. Data are presented as mean ± SD, $n = 4$ biologically independent animals.

deaths (Fig. 4d). 3 out of 4 rats died when TTX@Lipo containing 24.8 μg TTX was injected. Higher loadings of TTX (29.6 μg) were fatal. Nerve block from TTX@Lipo-Ph containing 25.3 μg TTX (Fig. 4b) lasted 36.9 ± 4.6 h (~2-fold the duration obtained with 20.4 μg TTX in

TTX@Lipo), there was no contralateral block (Fig. 4c), and there were no animal deaths (Fig. 4d). Nerve block from TTX@Lipo-Ph 32.1 μg TTX (Fig. 4b) lasted 57.1 ± 11.6 h even though this dose was higher than a dose that was uniformly fatal in animals receiving TTX@Lipo (29.6 μg

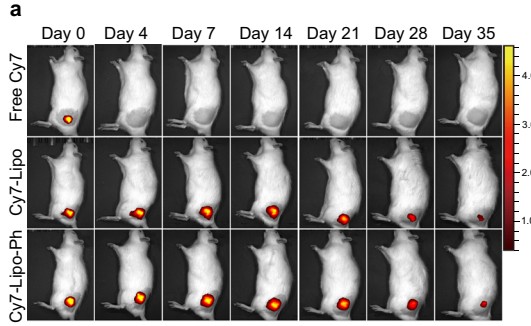
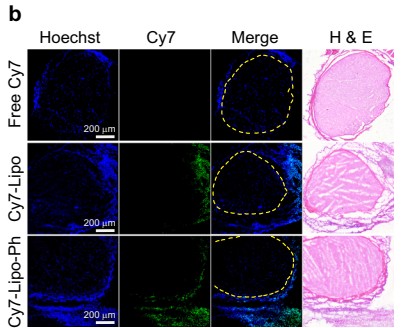

**Fig. 5 | Tissue distribution of aromatized liposomes. a** Representative fluorescent in vivo imaging of rats injected with different Cyanine 7 (Cy7)-labeled formulations. Fluorescence intensity is represented as radiant efficiency. Day 0 = early on day of injection. **b** Representative fluorescent confocal photomicrographs 24 h after injection of different Cy7-labeled formulations, with corresponding hematoxylin-eosin-stained (H & E) sections. Scale bar: 200 µm. Blue: Hoechst 33342, indicating cell nuclei; Green: Cy7, indicating dye-labeled liposomes. The yellow dotted line indicates the nerve perimeter. Each experiment was repeated in 4 biologically independent animals, yielding similar results.

TTX). TTX@Lipo-Ph containing 37.5 µg TTX enabled nerve block lasting 64.5 ± 8.1 h. Increasing the TTX loading to 44.2 µg further increased the duration of block to 70.7 ± 9.8 h (Supplementary Table 5). Even at that loading, there were no animal deaths, and contralateral block (lasting 0.9 ± 0.4 h) only occurred in 50% of animals (lower than the percentage with 20.4 µg TTX in TTX@Lipo). These data were consistent with the slower release kinetics of TTX@Lipo-Ph and demonstrated that liposome aromatization resulted in prolonged block and reduced toxicity.

To compare the covalently aromatized liposomes with liposomes where aromatic molecules were physically encapsulated, we injected ICG + TTX@Lipo containing 24.4 µg TTX at the sciatic nerve. Sensory nerve block lasted 26.9 ± 5.5 h (Fig. 4b), which was 1.35-fold longer than that with 20.4 µg TTX in TTX@Lipo, but shorter than that from 25.3 µg TTX in TTX@Lipo-Ph. Contralateral block (lasting 0.6 ± 0.2 h) occurred in 37.5% of animals (Fig. 4c; similar to that from TTX@Lipo) and there were no animal deaths (Fig. 4d). 2 of 4 animals administered 31 µg TTX in ICG + TTX@Lipo died. All animals receiving 38 µg TTX in ICG + TTX@Lipo died.

We then investigated whether the codelivery of encapsulated adjuvant compounds with aromatized liposomes can further enhance the nerve block duration. Previous studies have combined S1SCBs like TTX with adjuvants with α-adrenergic activity like epinephrine[26] and glucocorticoid receptor agonists like dexamethasone[28] for prolonged nerve blockade. TTX@Lipo-Ph containing 45.1 µg TTX combined with epinephrine (Epi, 3.6 µg) in the injectate (Epi + TTX@Lipo-Ph) and dexamethasone (Dex, 290 µg) in separate liposome (Dex@Lipo-Ph + TTX@Lipo-Ph) markedly increased the nerve block duration to 128.8 ± 22.3 h and 186.5 ± 23.8 h respectively (Fig. 4e). No contralateral block was observed in these groups. Prolonged duration local anesthesia (PDLA) lasting weeks is desirable for treating prolonged pain like cancer pain.

In all animals, motor nerve blockade was assessed by a weight-bearing test (see animal studies section in the Methods). There was no statistically significant difference between the durations of sensory and motor nerve block (Supplementary Fig. 14).

**Localization and retention of liposomal formulations**
To assess whether liposome aromatization affected the tissue retention of liposomes, liposomes were covalently conjugated with Cy7 and injected at the sciatic nerve. Fluorescence images of rats were taken at different time points using an in vivo imaging system (IVIS) (Fig. 5a). While free Cy7 diffused away from the injection site within hours (Supplementary Fig. 15a), both unmodified liposomes (Cy7-Lipo) and aromatized liposomes (Cy7-Lipo-Ph) remained in place for at least 5 weeks (Supplementary Fig. 16). Fluorescent confocal microscopy

confirmed the localization of Cy7-conjugated liposomes (Fig. 5b) as well as free dyes (Supplementary Fig. 15b) in the connective tissue between muscles and nerves. These findings suggested that differences in anesthetic effect between formulations would be due to differences in drug release, not tissue retention.

**Cytotoxicity and tissue reaction**
The in vitro cytotoxicity to muscle and nerve of TTX formulations was evaluated in myoblast C2C12[29] and pheochromocytoma PC12[30] (Fig. 6a, b) cell lines, that are used to assess myo- and neuro-toxicity respectively. After 48 h of incubation with different TTX formulations (TTX, TTX@Lipo, ICG + TTX@Lipo and TTX@Lipo-Ph), cell viabilities as assessed by MTS assays were similar to that of PBS, indicating that liposomes containing Ph-DPPC did not cause cytotoxicity.

4 and 14 days after injection, sciatic nerves and surrounding tissues were harvested for histology. Liposomes could be seen at the sciatic nerve 4 days after injection (Fig. 6c). Small amounts of residual liposomes could still be identified at the injection site after 14 days (Supplementary Fig. 17), consistent with the IVIS results.

Local anesthetics, particularly prolonged duration local anesthetics, can be associated with myotoxicity and inflammation[31], although with site 1 sodium channel blockers tissue reaction is due to the vehicle, not the anesthetic[16]. Drug delivery systems themselves are known to cause inflammation[32] that can outlast the duration of nerve block and may enhance local anesthetic myotoxicity[33,34].

The sciatic nerves and surrounding tissues were harvested and sectioned, and tissue reaction was assessed by hematoxylin and eosin (H&E) staining. All liposomal formulations produced comparable degrees of inflammation (Fig. 6d). Inflammation was seen around the sciatic nerves, which is also commonly seen with a broad range of delivery systems[35]. All slides were scored for inflammation (0–4) and myotoxicity (0–6; see tissue harvesting and histology section in the Methods). There was no statistically significant difference in inflammation scores (1.5–3.0) between different liposomal formulations (Supplementary Table 6). No myotoxicity was observed for ICG + TTX@Lipo and TTX@Lipo-Ph 4 days after injection (Supplementary Table 7). Mild myotoxicity (0.5) was observed in rats injected with TTX@Lipo and TTX@Lipo-Ph 14 days after injection; there was no statistically significant difference in myotoxicity scores between these groups and the untreated groups. It should be noted that the inflammation and myotoxicity scores of these formulations are comparable or better than those of Exparel in the same animal model[17], suggesting the observed tissue reactions are acceptable for potential clinical use.

Sciatic nerves were stained with toluidine blue due to the low sensitivity of H&E staining for nerve damage (Fig. 6d). No nerve injury was observed in the free TTX, ICG + TTX@Lipo and TTX@Lipo-Ph

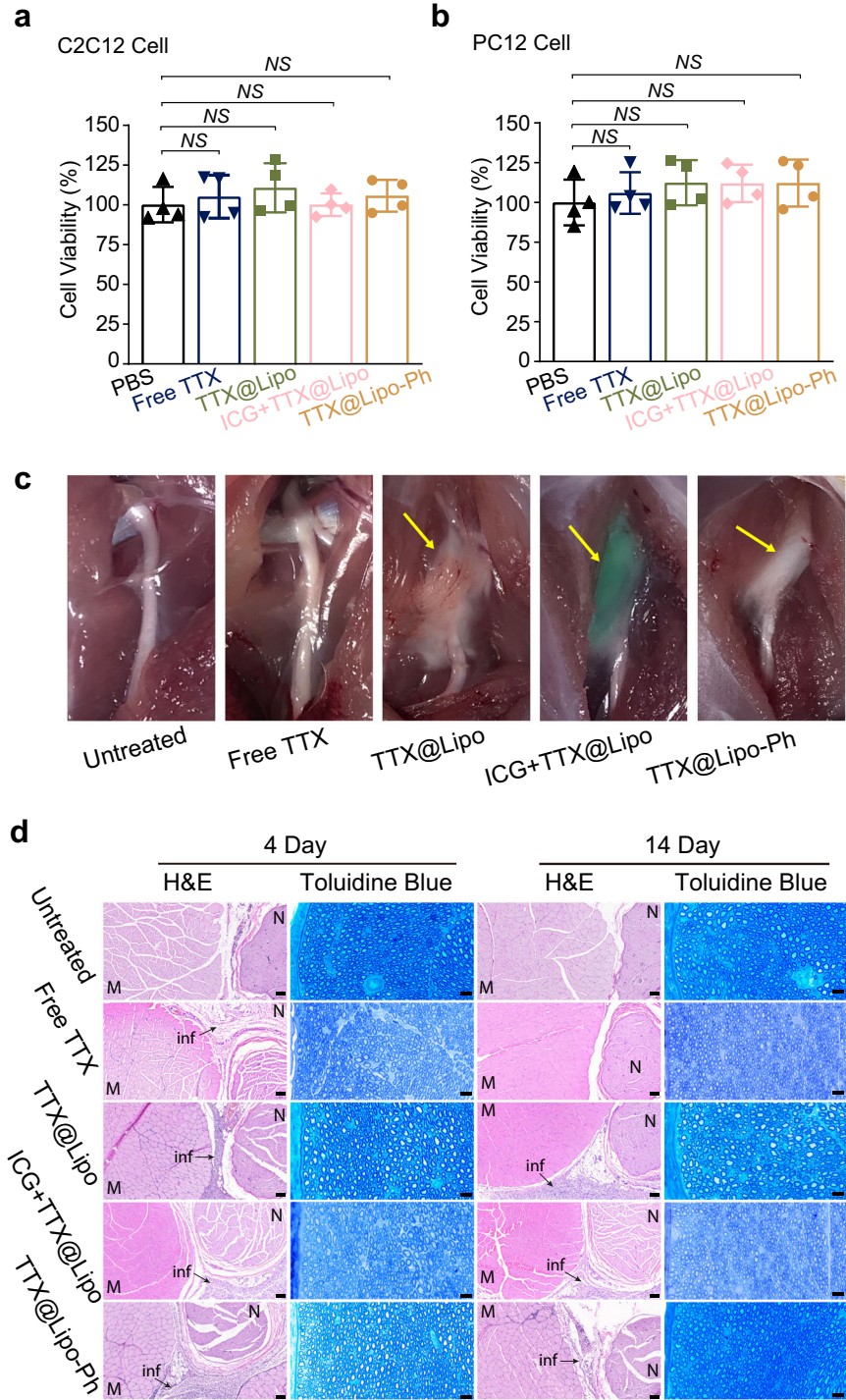

**Fig. 6 | Tissue reaction to TTX formulations. a, b** Cytotoxicity of formulations to C2C12 (**c**) and PC12 (**d**) cells. Black up-pointing triangle, navy down-point triangle, green square, pink diamond, and orange circle represent PBS, free TTX, liposomes encapsulating TTX (TTX@Lipo), liposomes encapsulating TTX and ICG (ICG + TTX@Lipo) and aromatized liposomes encapsulating TTX (TTX@Lipo-Ph), respectively. Data are presented as mean ± SD, *n* = 4 independent experiments. Statistical analysis was performed using one-way ANOVA with a Tukey post hoc test. **c** Representative photographs of the sites of injection upon dissection 4 days after injection. The green color is due to ICG. Arrows indicate the locations of formulations. **d** Representative H&E-stained sections of nerves and surrounding tissues, and toluidine blue-stained section of nerves. The scale bar is 100 μm for H&E-stained sections; 20 μm for toluidine blue.

groups. Small areas of sub-perineurial damage were observed in the TTX@Lipo group. There is a substantial track record of perineurial safety of the materials used here, and of TTX itself[18,27,36]. Moreover, other groups (TTX@Lipo-Ph, ICG + TTX@Lipo) that incorporated all the components of TTX@Lipo did not show injury. Therefore, it is likely that the injury was related to trauma from injection (the injections are done percutaneously, i.e., not under direct observation), or

injury at the time of dissection, rather than the composition of matter of TTX@Lipo.

## Discussion

Here, we report the improvement of sustained drug release by the aromatization of the inner aspect of liposomal lipid bilayers. Aromatized liposomes increased the drug loading and considerably

decreased the release of payloads with different water solubilities and molecular weights. These changes had an impact in vivo: aromatization prolonged the duration of local anesthesia from TTX liposomes to more than 3 days, and curtailed systemic toxicity.

Aromatic groups were incorporated within the lipid bilayers of liposomes to target the rate-limiting step in the passive diffusion of molecules from liposomes[12]. There are a number of mechanisms by which this might occur. π–π stacking interactions non-covalent crosslinked adjacent phospholipids[37], decreased lateral motion of lipids within the liposomal membranes, stabilized lipid bilayers, decreased their fluidity and permeability. The rigid ring structures of aromatic groups may reduce the fluidity and permeability of lipid bilayers as observed in some bacteria[38]. The aromatized liposomes may find broad application in the encapsulation and release of a broad range of drugs, including chemotherapeutics, macromolecular drugs, and proteins drugs. The release kinetics suggested the reduced release of small molecules due to aromatization likely correlates to multiple factors. Neither molecular weight nor hydrophilicity correlated strongly with the degree of reduction. The reduced release of macromolecules as result of aromatization correlated with their molecular weight and hydrophilicity.

There were constraints on the effect of aromatization on the loading of liposomes and the resulting release kinetics. While the addition of one ring enhanced performance, a second ring (coumarin) had minimal effect on liposome performance, and greatly increased viscosity—which would adversely affect removal of free drug from the formulation, and injectability in the clinical application. Addition of a bulky three-member ring (DBCO) actually impaired performance. That impairment may be due to the bulky group interfering with the packing of hydrocarbon chains and creating free space within lipid bilayers[20], which could increase the permeability of liposomes[21]. The DBCO group was conjugated to the acyl chain via two amide bonds, which may also make the liposome leakier. It remains to be seen whether further hydrophobic non-bulky modifications of aromatic groups would further enhance liposome performance.

Conventional anesthetics are short in duration and can have associated neurotoxicity and myotoxicity[31]. Limited therapeutic alternatives in pain management have produced an overreliance on opioid anesthetics worldwide[39]. A long-acting formulation of bupivacaine liposome (Exparel)[40] has been approved for clinical use in local anesthesia[2]. However, a recent study revealed that Exparel did not demonstrate significant pain relief compared to standard bupivacaine in 74.58% of randomized clinical trials[41]. Developing new formulations for prolonged non-addictive pain relief has long been of research and clinical interest[31]. Here we have described a formulation that provides three days of nerve block from a single injection. Addition of dexamethasone further extended the duration of analgesia to over a week. One important consideration in that context is that use in larger animals (such as humans) will allow for the use of larger doses, enabling longer blocks. There would also be less systemic toxicity because it tracks relatively linearly with the compound's volume of distribution (i.e., the animal's size) while the duration of nerve block does not. Nerve block lasting 2-3 days would cover the duration of most postoperative pain. Longer durations would be useful for severe localized chronic pain like cancer pain.

The majority of local anesthetics cause both sensory and motor nerve blockade; this is true of local anesthetic sustained release systems as well. In most cases, the treatment of pain would justify the accompanying immobility, and one might not want to be moving a painful body part. Moreover, the prolonged analgesia could obviate the need for opioid use. However, if the motor block greatly exceeded the duration of sensory block, or it the duration of block was very long, one might have to consider whether alternative approaches would be preferable.

Unlike perhaps most reports on the use of liposomes, the clinical application model used here involved local administration and effect, not systemic (intravenous). In that context, the neurobehavioral deficits in the injected extremity are a relevant bioassay of local drug kinetics and showed that the aromatized liposomes greatly enhanced the duration of nerve block. Similarly, deficits in the contralateral (uninjected) extremity are a reliable bioassay of systemic drug distribution[26], i.e., of the presence of TTX in the blood. Aromatization clearly affected the systemic pharmacokinetics of TTX, reducing toxicity and allowing larger doses and longer durations of effect.

In conclusion, aromatized liposomes increased the drug loading and reduced the release of encapsulated compounds compared to conventional liposomes. In vivo, they greatly prolonged the duration of local anesthesia from the small hydrophilic and charged molecule TTX and mitigated the systemic toxicity.

## Methods
### Materials
Unless otherwise specified, the number in parentheses following the names of reagents is the purity, provided by the manufacturer. 16-bromohexadecanoic acid (99%), acetyl chloride (99%), anhydrous methanol (99.8%), phenol (99%), potassium carbonate (99%), anhydrous acetonitrile (99.8%), anhydrous tetrahydrofuran (99.9%), hydrochloric acid (37%), 2,6-dichlorobenzoyl chloride (99%), 1-methylimidazole (99%), cholesterol (99%), sulforhodamine B (SRho, 75%), bupivacaine hydrochloride (99%), and doxorubicin hydrochloride (98%) were purchased from Sigma-Aldrich (St. Louis, MO, USA). Albumin–fluorescein isothiocyanate conjugate (FITC-Ab) was purchased from Sigma-Aldrich (St. Louis, MO, USA) with the catalog number as A9771. Cyanine 7 carboxylic acid was purchased from Lumiprobe Corporation (Hallandale Beach, FL, USA) with a catalog number as 15090. Polyethylene glycol with a molecular weight of 1000 (PEG1k, 95%) and polyethylene glycol with a molecular weight of 10000 (PEG10k, 95%) were purchased from JenKem Technology USA Inc (Plano, TX, USA). 16:0 DPPC (99%), 18:1 DOPC (99%), 18:0 DSPG (99%), 16:0 lyso PC (99%), Cy7-DOPC (99%) were purchased from Avanti Polar Lipids (Alabaster, AL, USA). Tetrodotoxin (TTX, 98%) was obtained from Abcam (Cambridge, MA, USA). TTX ELISA kits were purchased from REAGEN LLC (San Diego, CA, USA). Dulbecco's phosphate buffered saline (PBS) were purchased from Thermo Fisher Scientific (Waltham, MA, USA).

### Synthesis of aromatic group-modified phospholipids
Synthetic procedures are described in detail in Supplementary Methods in the supplementary information file.

### Liposome preparation and characterization
Liposomes were produced by the thin-film hydration technique, followed by subsequent agitation and freeze-thaw cycles[18,36]. A lipid mix containing Ph-DPPC, DOPC, DSPG, and cholesterol in a 3:3:2:3 molar ratio was dissolved in a solution of 90% chloroform and 10% methanol. This mixture underwent evaporation at low pressure, after which the lipid content was reconstituted in tert-butanol and subjected to freeze-drying. The resulting lipid cake was then rehydrated with different payloads, including Sulforhodamine B (SRho), tetrodotoxin (TTX), polyethylene glycol (PEG), and albumin–fluorescein isothiocyanate conjugate (FITC-Ab) in PBS buffer (pH=7.4) or bupivacaine hydrochloride (Bup), or doxorubicin hydrochloride (Dox) in saline. After 10 cycles of freezing and thawing, the liposomal solution was subjected to a 48-h dialysis against either PBS or water. The molecular mass cut-off of the dialysis devices was 1000 kDa (Spectra/Por™ Float-A-Lyzer™ G2 Dialysis Devices). The dialysis fluid was routinely replaced with fresh PBS or saline approximately every 12 h. To determine the drug and dye content across different formulations, liposomes were disrupted using

octyl-β-D-glucopyranoside (100 mM, twice the volume of liposomal solution) and analyzed.

## Instruments and characterization of materials

We utilized a Varian 400 MHz NMR spectrometer (Palo Alto, CA, USA) for ¹H and ¹³C NMR characterizations. The Zetasizer Pro from Malvern Panalytical (Westborough, MA, USA) was used to assess the liposomes' hydrodynamic diameter and zeta potential. To examine bupivacaine hydrochloride during the in vitro release tests, we employed an Agilent 1260 series HPLC system (Santa Clara, CA, USA), which incorporated a UV-vis detector and used a solvent mix of 40% acetonitrile (with 0.1% trifluoroacetic acid) and 60% water (also containing 0.1% trifluoroacetic acid). Molecular weight determinations were achieved using the Agilent Infinity lab LC/MSD XT single quadrupole mass spectrometer (Santa Clara, CA, USA). We measured viscosity utilizing a TA DHR-2 rheometer (New Castle, DE, USA), setting it at 1 Hz frequency, 1% shear strain, and maintaining the temperature at 25 °C. For differential scanning calorimetry (DSC) evaluations, we utilized the PerkinElmer DSC6000 (Waltham, MA, USA), operating under a nitrogen atmosphere. Samples were sealed with an aluminum crucible, and thermal transitions were measured and recorded between 20 °C and 80 °C, at a scanning rate of 1 °C/min.

## Cryo-EM imaging

Quantifoil copper EM grids (R2/1, 400 mesh) were glow-discharged for 30 s at 15 mA. This grid was loaded into the Vitrobot climate chamber at 100% humidity. 10 mg/mL liposome samples were prepared and 5 μL aliquots were manually dispensed onto the grids. Grids were blotted for 5 s with a wait time of 5 s and then plunged into liquid ethane for vitrification. Frozen grids were stored in liquid nitrogen until use. Cryo-EM grids were imaged on a Titan Krios II equipped with a Gatan K3 camera at 1.4 Å/pixel. 3.The Titan Krios II has an accelerating voltage of 300 kV. Image analysis and scaling were then done through ImageJ (Fiji) software.

## Partition coefficient quantification

We applied a shake-flask technique to measure the octanol–water partition coefficient (LogP) values[42]. Buffer solutions of 0.1 M ionic strength at three different pH values: 4.0 (citrate buffer), 7.4 (phosphate buffer), and 10.2 (carbonate buffer), were used to examine the ionized or neutral states of the payloads and their pH-dependent partition. It should be noted that we compare logP values of different payloads at pH 7.4, at which condition the payloads were encapsulated. Prior to testing, each buffer solution was saturated with octanol prior to analysis. After subjecting all mixtures to 5 min of vortexing, they were stirred at room temperature for 24 h, ensuring equilibrium and phase distribution. After equilibration was achieved, samples were analyzed either on an Agilent HPLC 1260 or on a plate reader (BioTek, Winooski, VT, USA). The HPLC used a solvent mix of 70% acetonitrile (containing 0.1% formic acid) and 30% water (containing 0.1% formic acid).

## Cell culture

Cell culture of C2C12 mouse myoblasts (American Type Culture Collection (ATCC, Catalog # CRL-1772, Manassas, VA, USA) and PC12 rat adrenal gland pheochromocytoma cells (ATCC, Catalog # CRL-1721, Manassas, VA, USA) were performed[18,43]. Prior to distribution, ATCC uses STR analysis to screen C2C12 and PC12 cell lines for authenticity and purity. Upon receipt of these cell lines and after plating, a morphology check by microscope on both C2C12 and PC12 is done to ensure authenticity. To summarize, C2C12 cells were cultured in DMEM with 20% FBS and 1% Penicillin Streptomycin. For differentiation into myotubules, these cells were seeded onto a 24-well plate at a density of 50,000 cells mL⁻¹ and then incubated for 10–14 days in DMEM containing 2% horse serum and 1% Penicillin Streptomycin.

PC12 cells were cultured in DMEM with 12.5% horse serum, 2.5% FBS, and 1% Penicillin Streptomycin. After seeding these cells onto a 24-well-plate, nerve growth factor at a concentration of 50 ng mL⁻¹ was introduced 24 h after seeding.

## Cell viability

For cytotoxicity assessment, liposomal formulations were directly added into the cell culture media and incubated in the media bathing the cells (i.e., in direct contact with them) in cell culture wells (lipid concentration: 20 mg/mL). After a 24 h incubation, cell viabilities were measured using the MTS assay. The survival rates were quantified as percentages of results in untreated cells.

## In vitro drug release

Cumulative release of small molecules (Sulforhodamine B, Tetrodotoxin, Bupivacaine hydrochloride, and doxorubicin hydrochloride) were performed by placing 200 μL of samples into a Slide-A-Lyzer MINI dialysis device (Thermo Fisher Scientific, Waltham, MA) with a 10,000 MW cut-off, further dialyzed with 14 mL release media and incubated at 37 °C on a platform shaker (New Brunswick Innova 40, 60 rpm)[18,43]. At predetermined intervals, the dialysis solution was exchanged with fresh, pre-warmed release media. The release media of Sulforhodamine B, Tetrodotoxin was PBS (pH = 7.4). The release media of Bupivacaine hydrochloride, and doxorubicin hydrochloride was physiological saline. The concentration of TTX in release media was quantified by an enzyme-linked immunosorbent assay (ELISA, Reagen LLC). The concentration of Sulforhodamine B in release media was determined by a plate reader (BioTek, Winooski, VT) with excitation and emission wavelengths of 560 nm and 580 nm. The concentration of bupivacaine hydrochloride (Bup) in release media was determined by high-performance liquid chromatography. The concentrations of SRho, TTX and Bup in release studies were 4 mM, 0.3 mM and 20 mM as reported previously[18,44].

The maximum cut-off of commercially available Slide-A-Lyzer MINI dialysis device was 20,000 MW. Thus, the cumulative release of macromolecules was performed by placing 500 μL of samples into a Float-A-Lyzer G2 dialysis devices (Spectrum Laboratories Inc, Piscataway, NJ) with a 1,000,000 MW cut-off, further dialyzed with 14 mL PBS and incubated at 37 °C on a platform shaker (New Brunswick Innova 40, 60 rpm). At predetermined intervals, the dialysis solution was exchanged with fresh, pre-warmed release media. The concentration of dye-conjugated macromolecules was determined by a plate reader (BioTek, Winooski, VT). The concentrations of SRho-PEG1k, SRho-PEG10k and FITC-Ab in release studies were 1 mM, 0.5 mM and 0.5 mM, respectively.

## Animal studies

Animal studies were conducted following a protocol (20-07-4222R) approved by the Boston Children's Hospital Animal Care and Use Committee, aligning with the guidelines of the International Association for the Study of Pain. The Ethics Committee of Boston Children's Hospital reviewed and approved the animal protocol. We used 10-week-old male Sprague–Dawley rats, with weights ranging from 350–400 g, purchased from Charles River Laboratories (Wilmington, MA, USA). They were housed in groups and maintained on a 12-h/12-h light/dark cycle, with light starting at 6:00 AM.

To perform sciatic nerve injections, a 23 G needle was utilized. Following brief anesthesia using isoflurane-oxygen, the needle was carefully positioned posteromedial to the greater trochanter, pointing in an anteromedial direction. Upon bone contact, the formulations were dispensed onto the sciatic nerve[18,27,43].

The starting point for dose selection for free TTX in the in vivo study was based on dose-response curves that we have reported[26], where 4–5 μg of TTX produced 1.5–2.5 h of sciatic nerve blockade, with a shorter duration of deficits in the contralateral extremity. The

starting dose of TTX in unmodified liposomes was adapted from our previous work[18], where 22 µg of TTX in liposomes produced effective sciatic nerve blockade lasting 17.2 h. A similar dose was used in the modified liposomes. Subsequent increases in dose were studied by incremental increases in dose-response curves, with dose escalation until systemic toxicity became dose-limiting.

Neurobehavioural evaluations were performed on both hind paws. Deficits in the right hindpaw, which were uninjected, served as a metric of systemic drug distribution. A modified hotplate testing[27,36] was employed to assess sensory nerve blockade. Each hind paw was sequentially exposed to a 56 °C hot plate (Stoelting, Wood Dale, IL, USA). The duration (in seconds) for which the rat allowed each hind-paw to remain on the hotplate was recorded as the thermal latency. A 2-s thermal latency was determined to be the baseline (indicating no nerve block), and a 12-s latency was used as the upper limit to avoid burns and/or hyperalgesia, at which time the hindpaw was removed from the hotplate. Nerve block was considered to be successful if latency exceeded 7 s. Each animal was given the hotplate test three times on each paw at each time point, and the average was used for subsequent data analysis.

To evaluate motor nerve block, we utilized a weight-bearing method to measure the hind paw's motor strength[27,36,45]. The rat was placed with one hind paw on a digital balance, allowing it to bear its own weight. The maximum weight it could bear without the ankle touching the balance was recorded. If the motor strength was less than 50% of the maximum strength, it was considered a successful motor block. The test was also repeated three times for each time point, and the average was used for subsequent data analysis.

Duration of sensory block was defined as the time needed for the thermal latency return to 7 s (a midpoint between the baseline and upper threshold)[18,36]. The motor block duration was calculated as the time for the motor strength to return to 50% of normal motor strength[27,36,45].

### Laser scanning confocal microscopy (LSCM) Imaging

Cy7-labeled liposomes were fabricated using Cy7-conjugated DOPC. The Cy7-conjugated DOPC was added to the lipid mixture and subsequently solubilized in a solution containing 90% chloroform and 10% methanol, with Cy7 concentration at 0.2 mg/mL[43,46]. Under reduced pressure, the solvent was removed. The lipid mixture was reconstituted in tert-butanol and subsequently freeze-dried. The resulting lipid cake was rehydrated in PBS buffer at pH = 7.4 to yield Cy7-labeled liposomes. Rats, briefly anesthetized with isoflurane-oxygen, received injections of 0.3 mL of different Cy7 labeled lipo-somal formulations. Sciatic nerves and adjacent tissues were collected and embedded into optimal cutting temperature (OCT) compound (VWR, Radnor, PA, USA), then cryopreserved at −20 °C. Using a Leica CM3050 S cryostat microtome (Wetzlar, Germany), tissue sections of 10 µm thickness were produced and mounted onto glass microscope slides. These sections were then treated with 4% paraformaldehyde for a duration of 20 min at ambient conditions, followed by three PBS (pH 7.4) washes. Hoechst 33342 was utilized to stain the nuclei. Subsequent imaging of these sections was undertaken using the Zeiss LSM 710 multi-photon confocal microscopy (Carl Zeiss AG, Oberkochen, Germany).

### In vivo imaging system (IVIS) imaging

Cy7-labeled liposomes were created using Cy7-conjugated DOPC. The Cy7-conjugated DOPC was integrated into the lipid mixture and sub-sequently solubilized in a solution containing 90% chloroform and 10% methanol, with Cy7 molar concentration at 0.2 mg/mL[43]. The solvent was removed by applying reduced pressure. The lipid mixture was reconstituted in tert-butanol and then freeze-dried. Rehydrating the lipid cake with pH 7.4 PBS buffer yield the Cy7-labeled liposomes. Under anesthesia using isoflurane-oxygen, the fur on the rats was

removed by shaving. They were then injected with 0.3 mL of different Cy7-labeled liposomes. Captured in vivo fluorescence images allowed for the measurement of fluorescence intensity at specific post-injection time intervals. Under isoflurane-oxygen anesthesia, the fluorescence intensity was measured using a Spectrum IVIS (Perki-nElmer, MA, USA). The imaging of the rats was recorded non-invasively[43]. For imaging process and recording, the 745 nm excita-tion filter and the 800 nm emission filter were applied. The Live Ima-ging® software associated with the IVIS was used for quantitative assessment.

### Tissue harvesting and histology

4- and 14-days post-injection, rats were euthanized, a decision based on our prior observations that these intervals are optimal for exam-ining both acute and chronic inflammation and myotoxicity[18,27,43]. The sciatic nerve and its adjacent tissues were collected for histological evaluations. A pathologist, blinded to the individual samples, assessed the samples for inflammation scores (on a scale of 0–4) and myo-toxicity (ranging between 0 and 6)[18,27,43].

The inflammation score was a subjective quantification of sever-ity, in which 0 indicated a normal state and 4 indicated severe inflammation. The myotoxicity score was determined based on two characteristics of local anesthetics' myotoxicity: the nuclear inter-nalization and regeneration of myocytes. Nuclear internalization was identified by myocytes having nuclei located away from their normal position at the periphery of the cell. Regeneration was identified by the presence of shrunken myocytes with basophilic cytoplasm. The scor-ing scale was as follows: 0 = normal; 1 = perifascicular internalization; 2 = deep internalization (more than five cell layers); 3 = perifascicular regeneration; 4 = deep tissue regeneration (more than five cell layers); 5 = hemifascicular regeneration; 6 = holofascicular regeneration.

To assess the formulations' potential neurotoxicity, the sciatic nerves were fixed using Karnovsky's KII solution, processed and embedded in Epon, and subsequently stained with toluidine blue. The stained samples were then assessed though optical microscope by a pathologist who was kept blind to individual samples.

### Statistical analysis

Statistical comparisons were performed using the unpaired two-tailed $t$-test unless stated otherwise. Thermal latency, inflammation and myotoxicity scores were reported as means and quartiles due to their ordinal or non-Gaussian character. Data are presented as means ± SD ($n = 4$) in release kinetics, cell work, neurobehavioral, and histology studies. To determine the statistically significant differences between the means of independent groups, the two-tailed $t$-test was used for two groups of data, and ANOVA with a Tukey post hoc test was used for multiple group comparisons. Details of sample size and statistical tests are specified in the figure legends. Data were considered statistically significant if $P < 0.05$ (****$P < 0.0001$, ***$P < 0.001$, **$P < 0.01$, *$P < 0.05$). All graphs were generated with GraphPad Prism 8.

### Reporting summary

Further information on research design is available in the Nature Portfolio Reporting Summary linked to this article.

## Data availability

The data that support the findings of this study are available within the article and its Supplementary Information files. The raw numbers for charts and graphs are available in the Source Data file whenever pos-sible. Source data are provided with this paper. The full image dataset is available from the corresponding author upon request.

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

## Acknowledgements

Support for this work was provided by NIH R35 GM131728 (to D.S.K.), the Anesthesia Research Distinguished Trailblazer Award (to Y.L. and W.Z.), the Technology Development Fund (to Y.L.), NIH K99/R00 GM141269 (to W.Z.), and NIH F32 CA257210 (to M.T.).

## Author contributions

Y.L. and D.S.K. conceived the project and designed the experiments. Y.L. carried out the experiments, analyzed data, performed the formal analysis, and wrote the original draft. T.J., A.L., X.D. carried experiments related to animal study. M.T. performed the histology assessment. R.S. carried tissue harvest and staining as well as Cryo-EM images. Y.Z., W.Z., R.Y. carried experiments related to chemical synthesis. D.W. carried experiments related to cytotoxicity study. X.L. carried experiments related to laser scanning confocal microscopy imaging. D.S.K. provided the funding sources, supervised the research, and revised the manuscript. The final manuscript was approved by all authors.

## Competing interests

The authors declare no competing interests.

## Ethics

Healthy adult male Sprague–Dawley rats weighing 400–450 g were purchased from Charles River Laboratories and cared for in accordance with protocols approved institutionally and nationally. Experiments were carried out in accordance with the Boston Children's Hospital Animal Use Guidelines and approved by Boston Children's Hospital's Animal Care and Use Committee.
