## [Peer Review File · Nature Communications]

Reviewers' Comments:

Reviewer #1:

Remarks to the Author:

The authors investigated different aromatized liposome formulations, with the hypothesis that aromatizing the phospholipids will prolong release of encapsulated small molecules from the liposomes through a number of different mechanisms. Experiments and results spanned from characterization of formulations, assessing them *in vitro*, to assessing them *in vivo* in an elegant nerve block assay, where contralateral nerve block serves as evidence of systemic drug distribution. Data are robust, proper controls are included to focus on different aspects of the liposome design, and each is clearly explained as to why it is included.

Significance- The ability to extend nerve block to three days (or more with added adjuvant) is significant, and they state that an advantage of this is avoidance of the use of opioids to manage post-surgical pain. There is no difference between sensory or motor nerve block (supp fig 14), and I do wonder that after a certain length of time the benefits of extended sensory nerve block would be outweighed by the issues associated with extended motor nerve block. This may be a point of discussion to consider.

Analytical approach- It is unclear why t-tests were used to analyze *in vitro* release data and not ANOVA that would allow for reporting of a main effect of each formulation. Reporting significance for one dose (fig 4b) makes sense as the non-aromatized formulations' effect did not last as long as TTX@lipo-Ph, but it is unclear why significance is only reported for one time point (fig 2e, fig 3, supp figs). In fig 3, it is not indicated at what time point the p-values indicate significance. N is also not indicated. The conclusions are supported by the data regardless, but some clarification would help.

Improvements- address questions regarding reporting of p values, selection of statistical tests, and minor edits to figure 3 caption.

For brief discussion- it is shown in supp fig 6 that micro-scale liposomes can be made <100 nm with extruding, but the question follows if the same effect of aromatization on loading and release kinetics would apply to nano-sized vesicles. This is an important factor to consider as systemically administered liposomes or extracellular vesicle based delivery systems would benefit from delayed release kinetics.

Minor edit- Supp table 3 says mm when it should say um

Reviewer #2:

Remarks to the Author:

This manuscript by Kohane and coworkers describes the preparation and *in vivo* analysis of chemically-modified aromatic liposomes for the delivery of local anaesthetic. Lipids are modified to feature aromatic groups (phenol, coumarin or DVCO), and their properties for encapsulating dyes or drugs are investigated. The authors show that adding the aromatic groups increases drug encapsulation (although modestly), and prolongs rate of release. This is tested *in vivo* in rat models showing prolonged anaesthetic effects as consequence of the aromatic liposomes. Overall this is a solid study and interesting body of work, but I don't think it meets the novelty criteria for publication in this journal. I recommend an alternative journal such as Journal of Controlled Release.

I additionally have the following comments:

- 1) The NMR spectra provided in the ESI are not integrated, this should be included as routine.
- 2) Only one small TEM image is provided of the liposomes which doesn't clearly confirm their structure (presence of an external membrane) and only shows two particles. I recommend providing an image which includes a larger particle sample size.
- 3) For the logP discussion, the values given in the text and from my understanding the values provided in the graph in Figure 3f, and Figure S11, appear to reference the values at pH 4 (from

Figure S10). However it clearly states in the methods that preparation of liposomes occurs in pH 7.4 PBS or saline, as well as release experiments. Therefore analysing this data in reference to pH 4 logP value doesn't make sense.

4) Figure 4e, the columns and legend do not match in terms of colours.

5) Figure 5 is very busy and very small, it is difficult to see individual details.

Reviewer #3:

Remarks to the Author:

The manuscript entitled "Aromatized liposomes for sustained drug delivery" demonstrated how new lipid chemistry can be used to alter the payload loading and delivery of various molecules having different MW and hydrophobicity (Fig.1 and supplementary data). The characterization of the lipid chemistry via NMR and the liposomes were clear. The focus was on payload encapsulation and release where the authors showed higher encapsulation and slower release (Fig. 2). The release of multiple molecules was investigated with the optimal formulation (Fig. 3). In vivo application of nerve block demonstrated extended effects (Fig.4). No nerve damage was measured (Fig.5). Overall the novel lipids showed a significant effect and demonstrated an important effect in vivo to extend nerve block.

Overall the article is novel. A few comments that may be addressed.

1. The particles measured were 1 micron was this optimal?

2. The toxicity of the supplementary Fig, 9 shows high toxicity but this did not effect the in vivo results. please comment.

3. The release of the various molecules was up to 20-30% of that encapsulated by 100 hours. Yet, the nerve block ended around the same time. Please comment on the potential difference on in vivo release rate and nerve block. For example, data of release in the presence of serum may elucidate these effects but is not necessary.

4. Fig. 4d shows mortality. Please indicate the time point that this was measured in the figure caption.

Reviewer #1's Comments

“The authors investigated different aromatized liposome formulations, with the hypothesis that aromatizing the phospholipids will prolong release of encapsulated small molecules from the liposomes through a number of different mechanisms. Experiments and results spanned from characterization of formulations, assessing them in vitro, to assessing them in vivo in an elegant nerve block assay, where contralateral nerve block serves as evidence of systemic drug distribution. Data are robust, proper controls are included to focus on different aspects of the liposome design, and each is clearly explained as to why it is included.”

Author Response: We thank the reviewer for this very positive review.

1. *“Significance- The ability to extend nerve block to three days (or more with added adjuvant) is significant, and they state that an advantage of this is avoidance of the use of opioids to manage post-surgical pain. There is no difference between sensory or motor nerve block (supp fig 14), and I do wonder that after a certain length of time the benefits of extended sensory nerve block would be outweighed by the issues associated with extended motor nerve block. This may be a point of discussion to consider.”*

Author response: Local anesthetics induce both motor and sensory nerve blocks. That is also true for sustained release systems of local anesthetics. In this work, the durations were not statistically significantly different. In response to the reviewer's comment, we have clarified that point in the manuscript, and added the following to the discussion:

“The majority of local anesthetics cause both sensory and motor nerve blockade; this is true of local anesthetic sustained release systems as well. In most cases, the treatment of pain would justify the accompanying immobility, and one might not want to be moving a painful body part. Moreover, the prolonged analgesia could obviate the need for opioid use. However, if the motor block greatly exceeded the duration of sensory block, or if the duration of block was very long, one might have to consider whether alternative approaches would be preferable.”

2. *“Analytical approach- It is unclear why t-tests were used to analyze in vitro release data and not ANOVA that would allow for reporting of a main effect of each formulation. Reporting significance for one dose (fig 4b) makes sense as the non-aromatized formulations' effect did not last as long as TTX@lipo-Ph, but it unclear why significance is only reported for one time point (fig 2e, fig 3, supp figs). In fig 3, it is not indicated at what time point the p-values indicate significance. N is also not indicated. The conclusions are supported by the data regardless, but some clarification would help.”*

Author response: We thank the reviewer for pointing this out and have revised the statistical analysis as suggested. To determine the statistically significant differences between the means of independent groups, the two-tailed t-test was used to compare two groups of data, and ANOVA was used for multiple group comparisons (with a Tukey post hoc test). Details of sample size and statistical tests are specified in the figure legends. We have revised the statistical analysis part in the method section accordingly. Here, we reported the significance at the 168-h time point as examples to compare the effect of aromatization on the release of payloads. We could have used any time point for the same purpose, and it was not clear what would be gained by reporting the significance at every time point, as it would not change the conclusion.

3. "Improvements- address questions regarding reporting of p values, selection of statistical tests, and minor edits to figure 3 caption."

Author response: We thank the reviewers for the suggestion. We have revised the figures captions to specify the details of sample size and statistical tests. In Figure 3, for example, data are the mean \pm s.d., n = 4 independent experiments. Statistical analysis was performed using one-way ANOVA with a Tukey post hoc test; p-values compare groups at 168 h.

4. "For brief discussion- it is shown in supp fig 6 that micro-scale liposomes can be made <100 nm with extruding, but the question follows if the same effect of aromatization on loading and release kinetics would apply to nano-sized vesicles. This is an important factor to consider as systemically administered liposomes or extracellular vesicle-based delivery systems would benefit from delayed release kinetics."

Author response: We appreciate the reviewer's comments, especially considering the potential systemic administration of aromatized liposomes, where sustained drug release would be highly beneficial.

We have conducted additional experiments to assess the effect of aromatization on nanoscale liposomes (100 nm). For this purpose, we prepared nanoscale aromatized and non-modified liposomes, maintaining the same composition as their micro-scale counterparts. We performed release kinetic experiments utilizing a representative payload, Sulforhodamine B (SRho). We found similar effects observed in micro-scale liposomes: aromatization indeed increases drug loading and decreases the release rate of payloads, even in the nano-sized vehicles. We have incorporated the new results into the revised manuscript (Supplementary Fig 6 e, f).

Fig. 1. Characterization of nanoscale liposomes. a, SRho loading in different nanoscale liposomes. b, Cumulative release of SRho from different formulations at 37 °C. In a and b, data are the mean \pm s.d., n = 4 independent experiments. Statistical analysis was performed using the two-tailed t-test. In b, the p-values compares groups at 96 h.

5. Minor edit- Supp table 3 says mm when it should say um

Author response: We thank the reviewer for paying attention to details and pointing out this error. We have now corrected this typographical error in the revised version of the table as per your suggestion.

Response to Reviewer #2's Comments

“This manuscript by Kohane and coworkers describes the preparation and in vivo analysis of chemically-modified aromatic liposomes for the delivery of local anaesthetic. Lipids are modified to feature aromatic groups (phenol, coumarin or DBCO), and their properties for encapsulating dyes or drugs are investigated. The authors show that adding the aromatic groups increases drug encapsulation (although modestly), and prolongs rate of release. This is tested in vivo in rat models showing prolonged anaesthetic effects as consequence of the aromatic liposomes. Overall this is a solid study and interesting body of work, but I don't think it meets the novelty criteria for publication in this journal. I recommend an alternative journal such as Journal of Controlled Release.”

Author response: We thank the reviewers for the positive comments on our experimental design and in vitro and in vivo studies. The reviewer's view that the effect on loading is modest is subjective; it was significant enough that between the increased loading and the perhaps more important effect of slowing release, there was a marked effect on duration of effect in vivo. We feel this work is very novel and demonstrates a substantial improvement in liposomal drug delivery.

1. *“The NMR spectra provided in the ESI are not integrated, this should be included as routine.”*

Author response: We appreciate the reviewer's suggestion regarding the integration of NMR spectra in the SI. We have updated the SI to include integration in every NMR spectrum.

2. *“Only one small TEM image is provided of the liposomes which doesn't clearly confirm their structure (presence of an external membrane) and only shows two particles. I recommend providing an image which includes a larger particle sample size.”*

Author response: We thank the reviewer for the valuable advice. We have now re-characterized the liposomes using cryogenic electron microscopy (cryo-EM). This technique offers more detailed visualization and clarity in defining the morphology and lamellarity of the liposomes. We have included representative images containing multiple liposomal vesicles in the revised manuscript (Fig 2a). Our updated cryo-EM confirm that the aromatized liposomes exhibit a mixture of unilamellar and multivesicular spherical vesicles, similar in size to conventional liposomes composed of natural lipids.

Conventional liposomes

Aromatized liposomes

3. “For the logP discussion, the values given in the text and from my understanding the values provided in the graph in Figure 3f, and Figure S11, appear to reference the values at pH 4 (from Figure S10). However it clearly states in the methods that preparation of liposomes occurs in pH 7.4 PBS or saline, as well as release experiments. Therefore analyzing this data in reference to pH 4 logP value doesn’t make sense.”

Author response: We thank the reviewer for pointing this out. The reviewer is correct: it would be more meaningful to compare compounds at a logP that related to the pH at which the compounds were encapsulated. (We had previously compared them at a pH where they were all in the same physical state, i.e., maximally water soluble). This changes the conclusion slightly. There is no correlation between logP at pH 7.4 and reduction in release rate (there was a moderate correlation with the data at pH 4). This is not a major difference – it just means that compounds with all logPs benefit similarly from aromatization). The correlation between logP and loading is much stronger at pH 7.4 than it was at pH 4.

The text has been changed as follows (changes are highlighted in red):

“Effect of liposome aromatization on different payloads. We expanded our study of the effect of liposome aromatization to include payload molecules with different characteristics such as hydrophilicity and molecular weight. The hydrophilicity of different payloads was measured by their octanol-water partition coefficients (LogP, **Supplementary Fig. 10**). In addition to SRho described above, we encapsulated the following small molecule compounds (**Fig. 3a**), using the same encapsulation and analytical techniques: tetrodotoxin (TTX, logP = 0.13), a hydrophilic ultrapotent local anesthetic; bupivacaine hydrochloride (Bup, logP = 1.24), an amphiphilic amino-amide local anesthetic in current clinical use; doxorubicin hydrochloride (Dox, logP = 0.33), a chemotherapeutic drug that has been used clinically in liposomes to treat cancer. **These are logPs at pH 7.4, the pH at which the compounds were encapsulated.**

Loading of small compounds was increased by 19-60% by aromatization of liposomes (**Supplementary Table 1**). **Although molecular weight did not correlate strongly with the degree of increase of drug loading ($R^2 = 0.32$), hydrophilicity showed good correlation with the degree of increase ($R^2 = 0.92$; **Supplementary Fig. 11**).** Release in the first 24 h was reduced by 30-60% by aromatization of liposomes (**Fig. 3b-d, Supplementary Table 2**). **Neither molecular weight (**Fig. 3e**; $R^2 = 0.01$) nor hydrophilicity (**Fig. 3f**; $R^2 = 0.00$) showed correlation with the degree of reduction.”.**

Fig. 3f has been changed as follows:

Supplementary Fig. 11b has been changed as follows:

4. *“Figure 4e, the columns and legend do not match in terms of colours.”*

Author response: We have now revised the legends in Figure 4e so that they correspond appropriately with the color of the columns. This correction should enhance clarity and facilitate better understanding of the figure.

5. *“Figure 5 is very busy and very small, it is difficult to see individual details.”*

Author response: In response to the reviewer’s comments, we have split Figure 5 into two separate figures, now designated as Figure 5 and Figure 6. This adjustment allows us to better display the details in each panel, making it easier for readers to understand the data.

Response to Reviewer #3's Comments

"The manuscript entitled "Aromatized liposomes for sustained drug delivery" demonstrated how new lipid chemistry can be used to alter the payload loading and delivery of various molecules having different MW and hydrophobicity (Fig.1 and supplementary data). The characterization of the lipid chemistry via NMR and the liposomes were clear. The focus was on payload encapsulation and release where the authors showed higher encapsulation and slower release (Fig. 2). The release of multiple molecules was investigated with the optimal formulation (Fig. 3). In vivo application of nerve block demonstrated extended effects (Fig.4). No nerve damage was measured (Fig.5). Overall the novel lipids showed a significant effect and demonstrated an important effect in vivo to extend nerve block. Overall the article is novel."

Author response: We thank the reviewers for this very positive review.

1. *"The particles measured were 1 micron was this optimal?"*

Author response: In response, we have produced and studied release from 100 nm liposomes. As shown in Supplementary Fig. 6, the release rate of SRho was substantially faster from nanoscale liposomes in comparison to their microscale counterparts. Whether that is optimal or not depends on the specific application and what is desired. In the case of local anesthesia with site 1 sodium channel blockers like TTX, where systemic toxicity could be an issue, slower release – which is generally obtained with larger particles – is likely better.

2. *"The toxicity of the supplementary Fig, 9 shows high toxicity but this did not effect the in vivo results. please comment."*

Author response: We thank the reviewer for the thoughtful comment regarding the toxicity shown in Supplementary Fig. 9 and its relation to our in vivo results.

To clarify, Supplementary Fig. 9 illustrates the cytotoxicity of liposomes containing physically encapsulated phenol (Ph@Lipo), where the "@" denotes physical encapsulation. This is distinct from our aromatized liposomes (Lipo-Ph), where the "-" symbolizes that aromatic groups are covalently conjugated onto lipids.

We have clarified this in the figure legend as follows:

"...liposomes loaded with aromatic molecules physically (@Lipo) or covalently (-Lipo)."

The cytotoxicity results in Supplementary Fig. 9 confirmed the toxicity of phenol molecules. Consequently, this formulation was not used for in vivo testing. For our in vivo experiments, we physically encapsulated ICG into the lipid bilayers to demonstrate the effect of physically encapsulated aromatic molecules on in vivo nerve block durations.

In summary, phenol, free or encapsulated in liposomes, was cytotoxic in vitro and so was not used in vivo. Phenol bound to lipids was not cytotoxic in vitro and was used in vivo, where it was not toxic.

3. *"The release of the various molecules was up to 20-30% of that encapsulated by 100 hours. Yet, the nerve block ended around the same time. Please comment on the potential difference on in vivo release rate and nerve block. For example, data of release in the presence of serum may elucidate these effects but is not necessary."*

Author response: For a sustained release system, the termination of in vivo nerve block does not necessarily indicate that TTX was completely depleted, only that the rate of release drops below a therapeutically effective level. This is commonplace in sustained release systems, especially local anesthetic ones. For example, in 1996 Berde and Langer reported bupivacaine-loaded PLGA microspheres that released drug for three weeks in infinite sink conditions. In vivo, block lasted 6-12 hours.

The fact that drug is still present and being eluted after block wears off is seen in the fact that dexamethasone, when added, was able to act on the released drug and prolong effect.

4. *“Fig. 4d shows mortality. Please indicate the time point that this was measured in the figure caption.”*

Author response: In the mortality results displayed in Figure 4d, deaths occurred within the first 0.5 hours post-injection in animals treated with free TTX, within 0.5-1.0 hours in animals treated with TTX@Lipo, and within 0.5 -1.5 hours in animals injected with TTX+ICG@Lipo. We have now revised the figure caption to include this information.

“Deaths occurred within the first 0.5 hours after injection in animals treated with free TTX, within 0.5-1.0 hours in animals treated with TTX@Lipo, and within 0.7-1.5 hours in animals injected with TTX+ICG@Lipo”

Reviewers' Comments:

Reviewer #1:

Remarks to the Author:

The revised manuscript benefits from additional discussion and data and is deemed suitable for publication.

Reviewer #2:

Remarks to the Author:

I am satisfied that the authors have addressed my comments sufficiently.

Reviewer #3:

Remarks to the Author:

All items were addressed. The manuscript is acceptable for publication. It demonstrates a novel application of liposome delivery systems.

REVIEWERS' COMMENTS

Reviewer #1 (Remarks to the Author):

The revised manuscript benefits from additional discussion and data and is deemed suitable for publication.

Author response: We are pleased to know that the reviewer finds the manuscript suitable for publication now. We thank the reviewer for reviewing our manuscript and for recognizing the improvement made.

Reviewer #2 (Remarks to the Author):

I am satisfied that the authors have addressed my comments sufficiently.

Author response: We are pleased to know that we have successfully addressed the reviewer's comments. We thank the reviewer once again for reviewing our manuscript and for enhancing the quality of our work.

Reviewer #3 (Remarks to the Author):

All items were addressed. The manuscript is acceptable for publication. It demonstrates a novel application of liposome delivery systems.

Author response: We are pleased to know that the reviewer finds the manuscript suitable for publication now. We thank the reviewer for reviewing our manuscript and for enhancing our manuscript's clarity and depth.